# Cell and molecular transitions during efficient dedifferentiation

**John ME Nichols[1], Vlatka Antolović[1], Jacob D Reich[1], Sophie Brameyer[2], Peggy Paschke[3], Jonathan R Chubb[1]***

[1]MRC Laboratory for Molecular Cell Biology and Department of Cell and Developmental Biology, University College London, London, United Kingdom; [2]Ludwig-Maximilians-University Munich, Martinsried, Germany; [3]CRUK Beatson Institute, Garscube Estate, Switchback Road, Bearsden, Glasgow, United Kingdom

**Abstract** Dedifferentiation is a critical response to tissue damage, yet is not well understood, even at a basic phenomenological level. Developing *Dictyostelium* cells undergo highly efficient dedifferentiation, completed by most cells within 24 hr. We use this rapid response to investigate the control features of dedifferentiation, combining single cell imaging with high temporal resolution transcriptomics. Gene expression during dedifferentiation was predominantly a simple reversal of developmental changes, with expression changes not following this pattern primarily associated with ribosome biogenesis. Mutation of genes induced early in dedifferentiation did not strongly perturb the reversal of development. This apparent robustness may arise from adaptability of cells: the relative temporal ordering of cell and molecular events was not absolute, suggesting cell programmes reach the same end using different mechanisms. In addition, although cells start from different fates, they rapidly converged on a single expression trajectory. These regulatory features may contribute to dedifferentiation responses during regeneration.

## Introduction

Dedifferentiation is the transition of a cell to a state characteristic of an earlier stage of development. This reversal of developmental programmes is a widespread response to tissue damage (*Merrell and Stanger, 2016*), allowing replenishment of stem cell populations, and has been implicated as a contributing process to cancer progression (*Friedmann-Morvinski and Verma, 2014*). Artificially triggered dedifferentiation is central to approaches to generate induced pluripotent stem cells (IPSCs) for tissue repair strategies (*Takahashi and Yamanaka, 2016*). Despite these important biological and clinical contexts, dedifferentiation is not well understood in any system – it would be fair to say that we do not even have an approximate conceptual framework for the main features of the process.

Previous studies have identified some candidate molecular players, including c-Jun (*Parkinson et al., 2008*), mTORC1 (*Willet et al., 2018*), histidine kinases (*Katoh et al., 2004*) and chromatin regulators such as CAF-1 (*Cheloufi et al., 2015*), although these are isolated with respect to any large-scale regulatory network. A recurring feature in IPSC studies is the hypothesis that dedifferentiation somehow recapitulates developmental intermediates, but in reverse (*Pasque et al., 2014*; *Cacchiarelli et al., 2015*). The support for these models is based upon a few developmental markers detected within reprogramming intermediates, rather than any formal cell type classification. The possibility of a stereotypical programme has been the subject of some debate, with some evidence for multiple gene expression trajectories, at least during IPSC derivation (*Stuart et al., 2019*). It is also not clear whether dedifferentiation should be considered as regulated in the sense of having checkpoints, monitoring the gradual activation of the necessary changes that make a stem cell.

*For correspondence:
j.chubb@ucl.ac.uk

**Competing interests:** The authors declare that no competing interests exist.

**Reviewing editor:** Richard Gomer,

These difficulties in understanding are for several reasons. Firstly, many of the characteristic models of dedifferentiation are slow – often taking days to weeks for effective return to the earlier developmental state. Dedifferentiation within a tissue context will be confounded by the mixed signatures from multiple cell types, and more often than not, by a lack of accessibility. Although IPSC generation in culture provides a more accessible model, the process is slow (weeks), a very small proportion of the starting population makes it back to the stem cell state (which means the cells one is interested in can be difficult to identify) and the process usually involves the forced expression of four transcription factors (TFs), two of which are proto-oncogenes (*Karagiannis and Yamanaka, 2014*).

To begin to formulate a framework for understanding the control of dedifferentiation, it would be useful to investigate a model that dedifferentiates effectively. Developing *Dictyostelium* cells can completely reverse their differentiation in around 24 hr (*Takeuchi and Sakai, 1971*; *Finney et al., 1987*; *Katoh et al., 2004*). The normal developmental programme of *Dictyostelium* is induced by starvation. Starving cells aggregate together into a multicellular mound, before differentiating into two major cell types – stalk and spore. Upon disaggregation and resupply of nutrients, at any time prior to terminal differentiation, the cells dedifferentiate, giving rise to cells that can feed, divide and develop as well as they could prior to the initial starvation process. Shortly after the onset of dedifferentiation, there is evidence for a critical decision phase. This phase – termed 'erasure' – corresponds to a loss of developmental memory (*Finney et al., 1979*). Prior to this phase, re-removal of nutrients causes rapid re-entry into the forward development process, an ability that is quickly lost as dedifferentiation proceeds. Initial microarray studies on the dedifferentiation process implied the overall gene expression programme is distinct from development (*Katoh et al., 2004*), going against the grain of the mammalian IPSC reprogramming studies that have argued for developmental recapitulation. Two mutants have been shown to affect aspects of dedifferentiation: the spontaneous mutant HI4 showed impairment in the loss of development-associated cell-cell adhesivity during dedifferentiation, although other features of the dedifferentiation response were unperturbed (*Finney et al., 1983*). Loss of the histidine kinase DhkA delayed the onset of cell population growth during dedifferentiation, although erasure, the initiation of DNA replication and overall dedifferentiation potential were not affected (*Katoh et al., 2004*).

In this study, we have carried out a detailed transcriptomic analysis of the dedifferentiation process in *Dictyostelium* and combined this with single cell imaging, to order the progression of gene expression and cell physiological changes occurring as cells dedifferentiate. Our data suggest that multiple phases of gene expression underlie the reversal of development, with a high degree of symmetry between the forward and reverse processes, but notable distinctions that can be explained by opposing biochemical processes required for nutrient rich or starvation conditions. Our overall analysis suggests a high degree of robustness to the dedifferentiation process, with strong mutations affecting cell growth still retaining relatively normal gene expression dynamics as cells return to the undifferentiated state.

## Results

### Genome scale features of dedifferentiation

To what extent do dedifferentiating cells retrace the gene expression trajectories they followed during development (*Figure 1A*)? Early microarray work on *Dictyostelium* dedifferentiation detected differences between the forward and reverse processes (*Katoh et al., 2004*). In contrast, mammalian cells undergoing induced reprogramming can display characteristics of specific developmental intermediates (*Pasque et al., 2014*; *Cacchiarelli et al., 2015*).

To characterise the gene expression transitions occurring during *Dictyostelium* dedifferentiation, we determined population level transcriptomes during a high temporal resolution time course of dedifferentiation. Dedifferentiation was initiated from the tipped mound phase of development (14 hr; *Figure 1B*), at which point cell type specialisation has commenced. Structures were gently disaggregated to single cells, which were inoculated into different types of growth media, or into phosphate buffer lacking nutrients. Cells were recovered at regular intervals from the dedifferentiation cultures, with RNAseq carried out on RNA extracted from these cells. We also prepared a reference forward developmental timecourse, to compare to dedifferentiation. Gene expression trajectories were summarised using principal component analysis (*Figure 1C*).

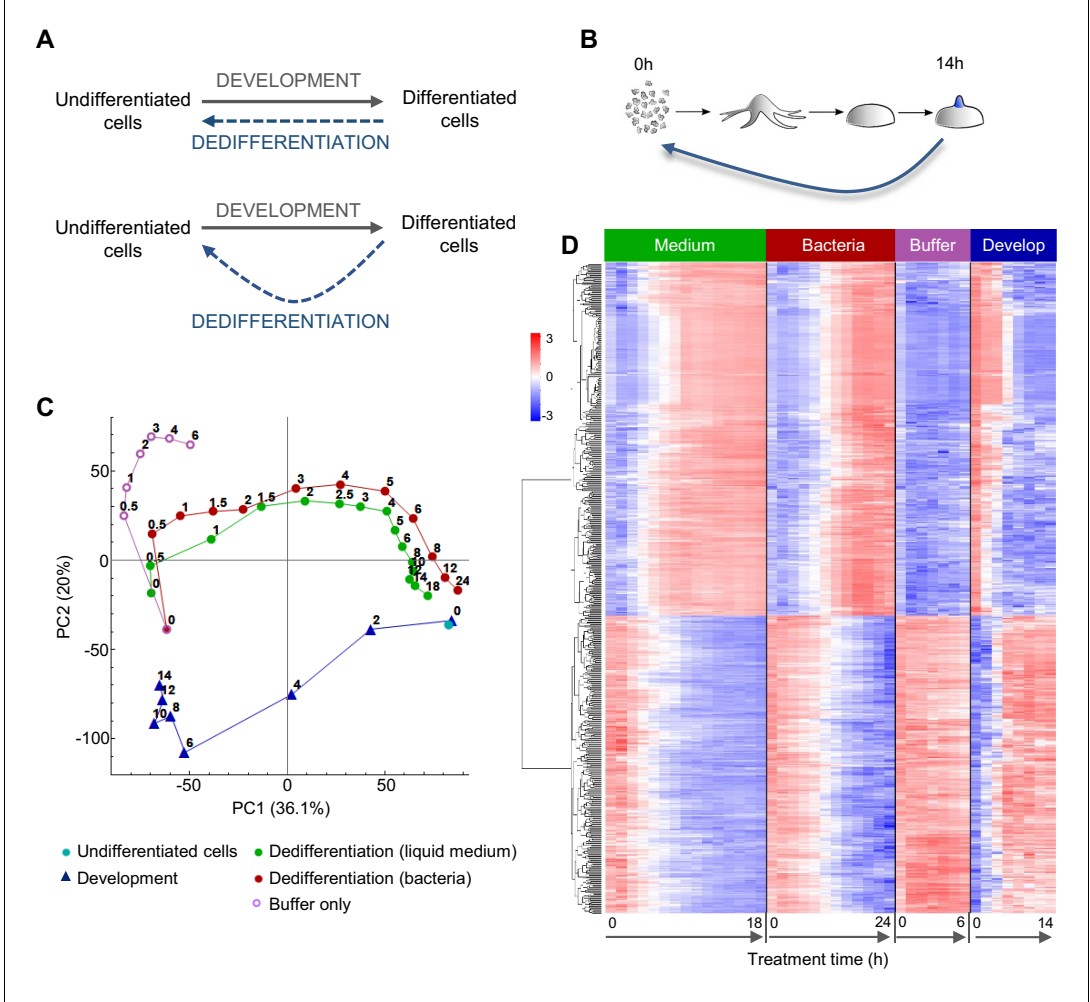

**Figure 1.** Comparing the gene expression trajectories of dedifferentiation and development. (A) Schematics of different dedifferentiation scenarios. Top: dedifferentiation is a simple reverse of forward development. Bottom: dedifferentiation visits distinct cell states during reversal. (B) *Dictyostelium* dedifferentiation is initiated by disaggregation of multicellular aggregates (after 14 hr of development) and transfer of the cells into nutrition (liquid medium or bacteria). (C) Dedifferentiation follows distinct gene expression trajectories compared to forward development. The figure shows principal component analysis (PCA) of RNAseq timecourse data from dedifferentiation in liquid medium and bacteria, forward development, mock dedifferentiation (buffer only) and a control undifferentiated sample captured alongside the dedifferentiation. Each point is the average of two replicates. Numbers on the plot represent time of sampling during dedifferentiation or development. (D) Hierarchical clustering of expression profiles of the 580 genes with highest contribution to the variance described by PC1. Expression changes over time during dedifferentiation in liquid medium and bacteria, buffer only and development are shown. Colour indication based on z-score of log2 read counts with high expression in red and low expression in blue.

The gene expression state of the dedifferentiating cells had almost completely returned to that of undifferentiated cells within 24 hr of dedifferentiation. Rapid recovery was observed whether dedifferentiation was induced using liquid (axenic) medium, or bacteria, as a food source. Both types of nutrition caused cells to reverse development along similar gene expression trajectories in the PCA space of the major principal components 1 and 2. For dedifferentiation under both nutritional conditions, most of the transcriptional changes along the PC1 axis had occurred after 6 hr. Although these trajectories had similar start and end points to the corresponding developmental stages, they diverged from the trajectory used by cells undergoing development, primarily with respect to PC2 values. Cells inoculated into nutrient free buffer after disaggregation showed a different path, which separated rapidly from the dedifferentiating trajectory (*Figure 1C*).

Although expression trajectories during dedifferentiation and development appear distinct, consistent with earlier microarray data (*Katoh et al., 2004*), analysis of the genes contributing most to

PC1 indicates the major changes in gene expression occurring during dedifferentiation are a straightforward reversal of development. The heat map in *Figure 1D* shows the results of an unbiased hierarchical clustering of 580 genes with the highest contribution to PC1 variance. The top cluster shows the genes activated during dedifferentiation. These are very similar between liquid medium and bacteria as a food source, although the bacterial response is slightly slower. These gene activation processes showed a strict reversal of the gene down-regulation occurring during development, although the timing was not strictly 'mirror image', insofar as 4 hr of dedifferentiation time did not correspond to 10 hr of developmental time (*Figure 1C*). In other words, gene expression milestones appear to be reached with different rates during the forward and reverse processes. Similar conclusions are reached when considering the genes turned off during dedifferentiation (*Figure 1D*, bottom cluster). The process is a clear reversal of development as far as PC1 is concerned, although the reverse trajectory escapes the advanced developmental state faster than it was acquired. Cells disaggregated into non-nutrient buffer showed a strong impairment of the activations and repressions characteristic of dedifferentiation.

We then considered what cellular processes, based upon transcript signatures, are subject to change during dedifferentiation, and to what extent any changes are reversals of developmental changes. An early microarray study, carried out before high-level annotation of the *Dictyostelium* genome, found a complex mixture of functional enrichments (*Katoh et al., 2004*). The use of current transcriptomic measurements, combined with a richer genome annotation, is expected to provide more resolution. We carried out enrichment analysis (GO) on genes with strong contributions to PC1 of a simplified principal component space lacking buffer-treated cells (*Figure 2A*). Strong positive loadings to PC1 were enriched for terms related to translation and mitochondrial function (*Figure 2B* and *Figure 2—figure supplement 1A,B*). In particular, a large panel of ribosome proteins showed strong positive PC1 loadings, corresponding to strong expression towards the end of the dedifferentiation trajectory and at the onset of development (*Figure 2—figure supplement 1C, D*). Strong negative loadings for PC1 were spread between processes related to terminal differentiation, cell adhesion, cAMP signalling and autophagy (*Figure 2B* and *Figure 2—figure supplement 2*). These terms would be expected of developing cells, which are starving, adhering and signalling using cAMP as they differentiate, and correspondingly mark the final portion of the forward development trajectory, and the beginning of the dedifferentiation trajectories (*Figure 2—figure supplement 2A*). Considering the portion of the trajectories with the biggest change in PC1 values, we found that from the 2428 genes up-regulated during dedifferentiation under both conditions, 62% were down-regulated during forward development. Similarly, of the 2172 genes down-regulated during dedifferentiation under both conditions, 60% were up-regulated during development. These data suggest a strong symmetry between the dedifferentiation and development trajectories, with most genes showing changes (above a two fold threshold) directly reversing their behaviour between the two processes.

Despite this apparent symmetry, the forward and reverse trajectories are distinct along the PC2 axis. To identify signatures specific to dedifferentiation or development, we identified the genes that contribute to PC2 variance. Genes with strong positive loadings for PC2 showed a more complex representation of functional classes, but were enriched with respect to transcription, secretion and the proteasome (*Figure 2—figure supplement 3*). In particular, proteasomal components are expressed throughout the developmental trajectory, and only weakly detected in the majority of the dedifferentiation trajectory (*Figure 2—figure supplement 3D*). In contrast, negative loadings into PC2 are completely dominated by ribosome biogenesis (as opposed to general translation terms), with components expressed strongly throughout dedifferentiation, but not development (*Figure 2C* and *Figure 2—figure supplement 4*). Although PC2 indicates differences between the forward and reverse processes, we note that proteasome and ribosome biogenesis are effectively the converse of each other – one degrading, one allowing synthesis (*Figure 2C*). So although at the transcript level these differences in expression break the symmetry between dedifferentiation and development, consistent with earlier microarray data (*Katoh et al., 2004*), in terms of the final protein product, these transcript changes predict an effective reversal. This opposition of functions appears aligned to the differing needs of the cell under starvation or nutrient-rich conditions.

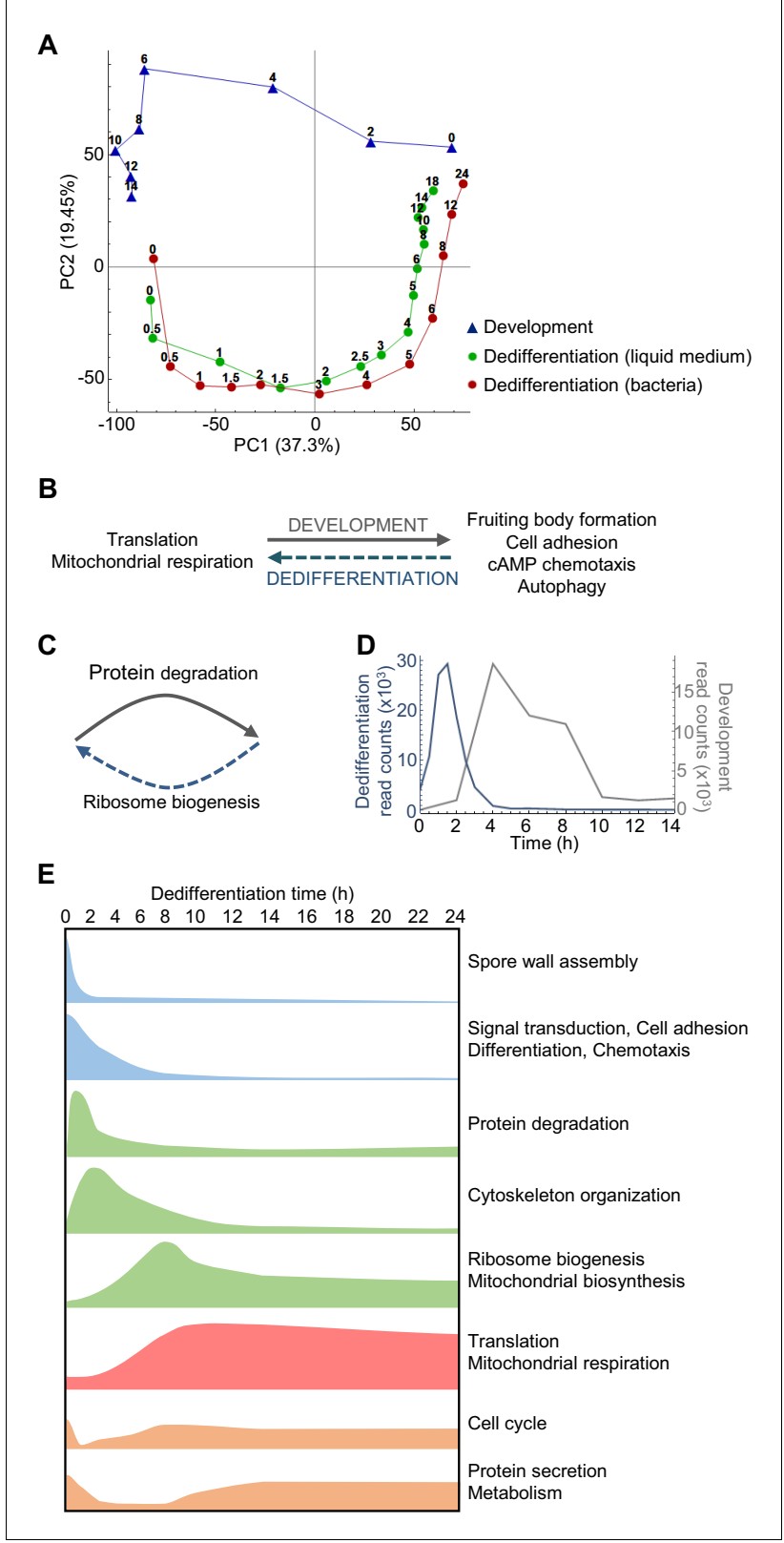

**Figure 2.** Overlapping and distinct transitions during dedifferentiation and development. (**A**) Simplified PCA: RNAseq timecourse data from forward development and dedifferentiation in liquid medium and bacteria. (**B**) Summary of gene expression transitions during dedifferentiation showing direct reversal of forward development. (**C**) Summary of gene expression transitions during dedifferentiation distinct from changes during development.
*Figure 2 continued on next page*

*Figure 2 continued*

(D) Rapid re-induction of a developmental gene during dedifferentiation. Dedifferentiation RNAseq counts in blue, with developmental counts in grey. (E) The major gene expression transitions of dedifferentiation. Data show time series of the changes in transcript read counts for different functional gene classes. Gene classes were determined by GO analysis of clusters of genes showing stereotypical temporal behaviour, identified by hierarchical clustering (refer to *Figure 2—figure supplement 5*, *6*, *7*, *8* and *9*).

The online version of this article includes the following figure supplement(s) for figure 2:

**Figure supplement 1.** Genes induced during dedifferentiation and repressed during development.
**Figure supplement 2.** Genes repressed during dedifferentiation and induced during development.
**Figure supplement 3.** Genes specific to development, not dedifferentiation.
**Figure supplement 4.** Genes specific to dedifferentiation, not development.
**Figure supplement 5.** Different classes of gene expression profile during dedifferentiation.
**Figure supplement 6.** Functional enrichment analysis of genes repressed during dedifferentiation.
**Figure supplement 7.** Functional enrichment analysis of genes transiently induced during dedifferentiation.
**Figure supplement 8.** Functional enrichment analysis of genes induced during dedifferentiation.
**Figure supplement 9.** Functional enrichment analysis of genes transiently repressed during dedifferentiation.

## Staging gene expression during dedifferentiation

A more specific example of developmental reversal is suggested by the strong reactivation of the *csaA* gene immediately after the induction of dedifferentiation (*Figure 2D*). The gene is normally expressed when cells aggregate during development, before the transcript declines in the aggregate, and encodes a cell surface protein to facilitate cell adhesion during development. We also identified other well-characterised aggregation-specific genes showing similar dynamic behaviour, such as *carA* and *pldB*. These effects are consistent with ideas that cells may revisit specific developmental states during dedifferentiation. Is this re-induction a general feature of aggregation genes? To address this question, we defined a panel of 402 aggregation-specific genes using the criteria that the mean expression over 4, 6 and 8 hr of development must be more than two fold greater than both the initial expression (0 and 2 hr) and the expression in the mound (10, 12 and 14 hr). We then compared these genes to those strongly induced during early dedifferentiation (defined as maximally induced by more than 50% in the first 2 hr). For dedifferentiation in bacteria, we found a 41% overlap (165/402 genes) between the dedifferentiation and aggregation genes. This overlap was compared to a simulated scenario with 402 genes sampled at random (10,000 times) from the entire genome. In the simulations, the median overlap between the dedifferentiation genes and the randomly sampled genes was found to be only 22%, with the 41% value never observed. This implies a strong enrichment of aggregation markers in the early dedifferentiation gene set. However, this effect was specific to the bacteria-based dedifferentiation, with liquid medium cultures showing an effect in the opposite direction, with only 15% overlap between dedifferentiation and aggregation-induced genes. We conclude that the re-induction phenomenon is not consistently a feature of dedifferentiation trajectories. An alternative explanation is that genes such as *csaA* and *carA* are repressed by strong cAMP signalling in the mound (*Van Haastert et al., 1992*; *Masaki et al., 2013*; *Cai et al., 2014*; *Corrigan and Chubb, 2014*). If this signalling is relieved (by disaggregation) then the repressive influence on transcription is removed, triggering re-induction.

Dedifferentiation is characterised by several distinctive classes of gene expression, occurring at different phases during the process (summarised in *Figure 2E*). These phases are revealed by hierarchical clustering of the RNAseq data (*Figure 2—figure supplement 5*). Early 'off' events (*Figure 2—figure supplement 6*) correspond to the down-regulation of genes encoding fruiting body components, in addition to other developmentally induced genes. Following this initial step are a series of transient inductions (*Figure 2—figure supplement 7*) starting with genes for protein degradation, then cytoskeletal components and regulators, before widespread induction of ribosome and mitochondrial biogenesis. After the transients come the stable long-term induction of translation and mitochondrial respiration components (*Figure 2—figure supplement 8*). A final grouping includes cell cycle genes (*Figure 2—figure supplement 9*), which are expressed initially (reflecting their role in development; *Muramoto and Chubb, 2008*) before switching off, then re-inducing as dedifferentiation proceeds.

**Table 1.** Gene expression in cells lacking candidate regulators of dedifferentiation.

Analysis of the gene expression phenotypes of cell lines mutated for candidate dedifferentiation regulators. Gene expression during dedifferentiation was assessed using a variety of methods, as indicated. For each assay one replicate was carried out unless stated otherwise.

| Gene | Description | Assay | Notes |
|---|---|---|---|
| *bzpS* | BZIP transcription factor | RNAseq | By PCA, slight delay at 2 hr. Other timepoints wild type. |
| *mybD* | MYB domain transcription factor | RNAseq | By PCA, slight delay at 2 hr. Other timepoints wild type. |
| *nfyA* | CCAAT-binding transcription factor | RNAseq | By PCA, developmental effect seen at 0 hr. Later time points wild type. |
| *DDB_G0269374* | Putative DNA binding protein | RNAseq | By PCA, very slight delay at 2 hr. Other timepoints wild type. |
| *DDB_G0272386* | F-box domain kelch repeat protein | RNAseq | By PCA, developmental effect seen at 0 hr. Later time points wild type. |
| *DDB_G0281091* | Acidic nuclear phosphoprotein | RNAseq | By PCA, all timepoints wild type. |
| *bzpI* | BZIP transcription factor | Act8 reporter expression by flow cytometry | Wild type |
| *eriA* | Putative RNAase III | Act8 reporter expression by flow cytometry | One clone retained larger than wild type Act8 reporter uninduced population. Not replicated in independent clone. |
| *fslN* | Frizzled and smoothened-like protein | Act8 reporter expression by flow cytometry | Wild type |
| *gbpD* | cGMP binding protein, RapGEF | Act8 reporter expression by flow cytometry | Wild type |
| *jcdA* | Jumonji domain transcription factor | Act8 reporter expression by flow cytometry | Wild type |
| *nfaA* | RasGAP | Act8 reporter expression by flow cytometry | Wild type |
| *ptpB* | Protein tyrosine phosphatase | Act8 reporter expression by flow cytometry | Wild type |
| *DDB_G0277531* | EGF-like domain protein | Act8 reporter expression by flow cytometry | Wild type |
| *ctnB* | Countin | Northern blot (*PCNA, csaA, hspE*) | *PCNA* and *hspE* wild type. Slightly increased *csaA* expression. |
| *gefAA* | LRR protein, RasGEF | Northern blot (*PCNA*) | Wild type |
| *gefS* | RasGEF | Northern blot (*PCNA, csaA, hspE*) | Wild type |
| *gtaN* | GATA transcription factor | Northern blot (*PCNA, csaA, hspE*) | *PCNA* wild type. Weak induction in *hspE*. Slight delay in down-regulation of *csaA*. |
| *krsB* | STE20 family protein kinase | Northern blot (*PCNA, csaA, hspE*) | Weak *PCNA* expression in one clone, not replicated in independent clone. *csaA* and *hspE* wild type. |
| *omt5* | o-methyltransferase | Northern blot (*PCNA, csaA, hspE, rpl15*) | Wild type |
| *pakE* | p21-activated kinase | Northern blot (*PCNA, csaA, hspE*) | Wild type |
| *rasG* | Ras GTPase | Northern blot (*PCNA, csaA, hspE, H2Bv1, sodC*) | Slight delay switching off *csaA*. Others wild type. |
| *sigB* | SrfA-induced gene | Northern blot (*PCNA*) | Wild type. |
| *sodC* | Superoxide dismutase | Northern blot (*PCNA, csaA, hspE, rpl15*) | Wild type |

*Table 1 continued on next page*

*Table 1 continued*

| Gene | Description | Assay | Notes |
|------|-------------|-------|-------|
| *tagA* | ABC transporter B family protein | Northern blot (*PCNA*) | Weak *PCNA* expression in one clone, not replicated in independent clone. |
| *xacB* | RacGEF, RacGAP | Northern blot (*PCNA, csaA, hspE*) | Wild type |
| *zakA* | Dual-specificity protein kinase | Northern blot (*PCNA, csaA, hspE, rpl15*) | Wild type |
| *DDB_G0268696* | Putative leucine zipper transcriptional regulator | Northern blot (*PCNA*) | Wild type |
| *DDB_G0269040* | IPT/TIG, EGF-like, C-type lectin domains | Northern blot (*PCNA*) | Weak *PCNA* expression in one clone, not replicated in independent clone. |
| *DDB_G0270436* | Putative RNA binding protein | Northern blot (*PCNA, csaA, hspE*) | Slight delay in down-regulation of *csaA*. Otherwise wild type. |
| *DDB_G0270480* | | Northern blot (*PCNA*) | Wild type |
| *DDB_G0272364* | EGF-like domain-containing protein | Northern blot (*PCNA*) | Wild type |
| *DDB_G0272434* | Notch/Crumbs orthologue | Northern blot (*PCNA*) | Wild type |
| *DDB_G0274177* | EGF-like domains | Northern blot (*PCNA*) | Wild type |
| *DDB_G0275621* | SET domain-containing protein | Northern blot (*PCNA, rpl15*) | Wild type |
| *DDB_G0276549* | Putative RapGAP | Northern blot (*PCNA*) | Wild type |
| *DDB_G0278193* | Orthologue of asparagine synthetase domain containing protein 1 | Northern blot (*PCNA*) | Wild type |
| *DDB_G0279851* | GCN5-related N-acetyltransferase | Northern blot (*PCNA, rpl15*) | Weak *PCNA* expression in one clone, not replicated in independent clone. *rpl15* wild type. |
| *DDB_G0280067* | Protein phosphatase 2C-related | Northern blot (*PCNA*) | Wild type. Bacterial grown cells due to liquid growth defect. |
| *DDB_G0283057* | Putative RapGAP | Northern blot (*PCNA, rpl15*) | Wild type |
| *DDB_G0288203* | Ifrd1 orthologue | Northern blot (*PCNA, rpl15*) | Weak *PCNA* expression in one clone, not replicated in independent clone. *rpl15* wild type. |
| *DDB_G0289907* | EGF-like, C-type lectin domains | Northern blot (*PCNA*) | Wild type |
| *DDB_G0292302* | F-box, Zn-finger protein | Northern blot (*PCNA, csaA, hspE, rpl15*) | Wild type |
| *DDB_G0293078* | Orthologue of FAM119B | Northern blot (*PCNA*) | Wild type |
| *DDB_G0293562* | LYAR zinc finger protein | Northern blot (*PCNA, csaA, hspE, rpl15*) | Wild type |
| *forG* | Formin | Northern blot (*PCNA, hspE, rpl15*), clonal recovery, RNAseq | Defect in expression of *PCNA* doublet upper band, observed in 3 independent clones. Slightly increased *hspE* at early timepoints. Defect in clonal recovery (4 replicates). Bacterial grown cells due to axenic defect. |
| *rasS* | Ras GTPase | Northern blot (*PCNA, hspE, rpl15*), clonal recovery | Defect in expression of *PCNA* doublet upper band (3 replicates). Slightly increased in *hspE* and *rpl15* at early timepoints. Bacterial grown cells due to axenic defect. |

## Genetic robustness of the dedifferentiation programme

More than 6000 genes changed their expression during dedifferentiation, under both conditions tested. To investigate the control of these genes, we identified six transcription factors (TFs) showing rapid induction in their expression during early dedifferentiation – BzpS, MybD, NfyA, DDB_G0272386, DDB_G0269374 and DDB_G0281091. To determine the genes regulated by these TFs, we mutated the coding sequences of their genes, then carried out RNAseq on the mutants during dedifferentiation. To our surprise, none of the mutants showed clear changes in gene expression compared to wild type (*Table 1*, examples shown in *Figure 3A*). This approach was clearly not a cost-effective strategy, so we next carried out an insertional mutagenesis screen (*Kuspa and Loomis, 1992*) to identify potential regulators of dedifferentiation. We generated an insertional library in an *act8*-mNeonGreen cell line. The *act8* gene (actin) is expressed in undifferentiated cells, but strongly repressed during development (*Tunnacliffe et al., 2018*). We enriched for mutants showing delayed induction of *act8*-mNeonGreen during dedifferentiation, by multiple iterations of flow sorting of low fluorescence cells. As an attempted proof of concept, we regenerated eight of the enriched mutants by homologous recombination in the *act8*-mNeonGreen cells. Unexpectedly, none of these mutants were consistently impaired in their ability to induce mNeonGreen during dedifferentiation (*Table 1*). We therefore considered a different approach, this time using a focussed screen, mutating genes with strong induction early during dedifferentiation, but minimal expression during development. Induction was primarily defined using hierarchical clustering, identifying genes with a transient increase of more than 2-fold during dedifferentiation. We focussed on genes with signalling and gene expression functions, and those overlapping with the insertional screen, favouring genes with clear induction in both bacterial and liquid culture-based dedifferentiation. The *dhkA* gene (*Katoh et al., 2004*) did not fit these criteria, showing a strong induction during development and a rapid loss of transcript early during dedifferentiation (*Figure 3—figure supplement 1A*).

Mutants were generated by homologous recombination- and then screened primarily by Northern blotting using a probe against the gene encoding PCNA. This marker is strongly induced around 5 hr after the onset of dedifferentiation, with many other cell cycle transcripts. To our surprise, none of the mutants showed a defect in *PCNA* induction (*Table 1*). Overall, these investigations suggested an apparent resilience of the dedifferentiation programme to genetic perturbation.

Key features of dedifferentiation are a return to cell division and growth. With this in mind, we considered the possibility that mutants of genes that are required for dedifferentiation would not be recovered, since they would be required for division and growth – indeed 17 mutants were not recovered using homologous recombination (See Appendix). To test this reasoning, we measured dedifferentiation-induced gene expression in the *forG* (formin) mutant, which has a growth defect during both bacterial and liquid culture (*Junemann et al., 2016*). *PCNA* expression was induced with relatively normal timing in *forG-* cells (*Figure 3B*), however induction of a longer transcript was impaired. The identity of the longer transcript was not clear, but may relate to a potential alternative promoter around 400 bp upstream of the normal transcription start site (*Figure 3—figure supplement 1B,C*). Clonal recovery on bacterial lawns of dedifferentiating *forG-* cells was reproducibly lower than that for wild types (*Figure 3—figure supplement 1D*), perhaps due to the strong phagocytosis defect of the mutants. Despite the slightly perturbed induction of *PCNA* and weakened clonal survival, other markers of dedifferentiation timing, such as the *hspE* and *rpl15* genes, showed normal induction (*Figure 3B*, left panels). At the whole transcriptome level, the *forG-* cells showed a slight delay in the dedifferentiation response, although the starting position of the mutants in the PCA space was slightly shifted with respect to wild type (*Figure 3C*). The *rasS-* mutant also has a strong growth defect in liquid medium (*Chubb et al., 2000*; *Paschke et al., 2018*). As with the *forG-* cells, the *rasS-* mutants showed a weakened induction of *PCNA*, however the induction of *hspE* and *rpl15* was again similar to wild type, suggesting no genome-wide impairment of gene expression (*Figure 3B*, right panels). Unlike the *forG-* cells, dedifferentiating *rasS-* cells showed no defect in clonal recovery on bacterial lawns (*Figure 3—figure supplement 1E*), implying no absolute requirement for RasS in dedifferentiation. These observations indicate that defects in dedifferentiation gene expression can be detected in mutants with strong growth defects, however the gene expression dynamics still appear relatively robust.

Both the *forG-* and *rasS-* mutants have defects in macropinocytosis, which impair their growth in liquid medium. Even after 30 hr of dedifferentiation, when cell division is widespread in the culture

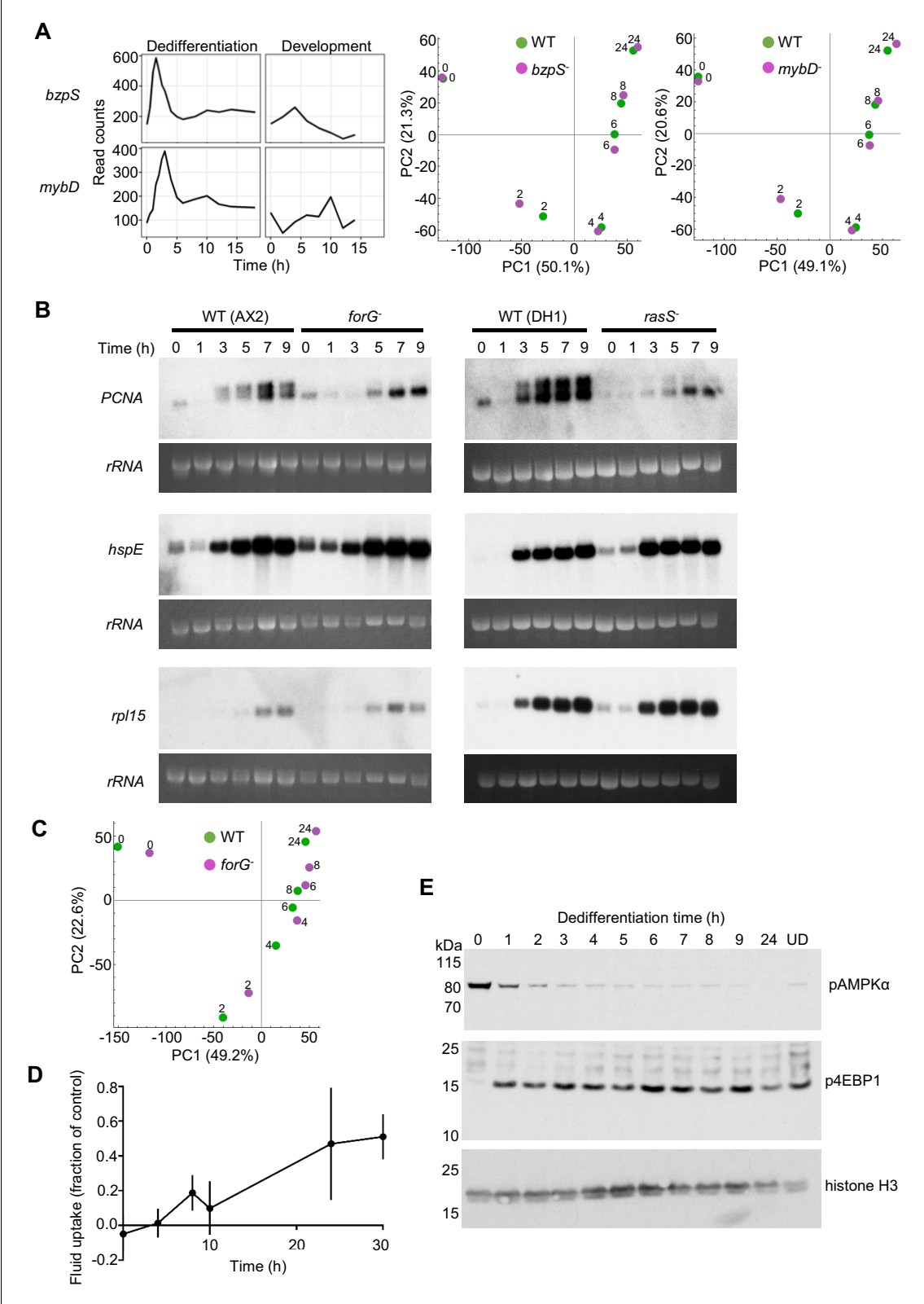

**Figure 3.** Molecular regulation of dedifferentiation. (**A**) Testing the importance of transcription factors expressed early in dedifferentiation. Expression of *bzpS* and *mybD* during dedifferentiation in liquid medium alongside their developmental profiles (left). Right: PCA of transcriptome changes during dedifferentiation of wild type and *bzpS* and *mybD* mutants. See *Table 1* for details of additional mutants. (**B**) Disrupted gene expression during dedifferentiation in *forG⁻* and *rasS⁻* mutants. Northern blots of *PCNA, HspE* and *Rpl15* expression during dedifferentiation in wild-type, *forG-* and *rasS-*
*Figure 3 continued on next page*

*Figure 3 continued*

cells, with RNA loading indicated by 26S rRNA. *PCNA* blots representative of three experiments. (**C**) PCA of transcriptome changes during the dedifferentiation of *forG-* and wild-type cells. PCA carried out on the mean read counts of two biological replicates. (**D**) Onset of fluid uptake during dedifferentiation of wild-type (AX2) cells in liquid medium, measured as a fraction of the fluid uptake by undifferentiated cells. Data are the mean and SD of four replicates, except for 30 hr, with three replicates. (**E**) Rapid changes in phosphorylation of nutrient response markers during dedifferentiation in liquid culture. Phospho-western blotting of AMPKα and 4E-BP1 phosphorylation. UD = undifferentiated cell sample. Equal amounts of protein loaded, with histone H3 used as a standard (three replicates).

The online version of this article includes the following source data and figure supplement(s) for figure 3:

**Figure supplement 1.** Analysis of candidate regulators of dedifferentiation.
**Figure supplement 1—source data 1.** Fluid uptake and clonal recovery data.

and gene expression has returned to the undifferentiated state, the wild-type fluid uptake was only half the normal level of undifferentiated *Dictyostelium* cells. However, a small but reproducible increase in fluid uptake could be detected around 8 hr into dedifferentiation (*Figure 3D*). This modest amount of macropinocytosis during the early phase of dedifferentiation suggests the process is not strongly required for most of the gene expression changes occurring early on, and might explain why mutants with strong defects in fluid uptake do reasonably well in activating dedifferentiation gene expression. These observations hint that processes other than macropinocytosis drive the response to nutrition. In line with this possibility, standard sensors of the nutritional state of the cell (*Jaiswal and Kimmel, 2019*) show rapid responses to the induction of dedifferentiation (*Figure 3E*), with cells showing rapid loss of the phosphorylated form of AMPKα and rapid induction in the phosphorylation of the mTORC1 substrate 4E-BP1, long before macropinocytosis could be detected.

## Loose coupling between gene expression and cell physiology

As dedifferentiation appeared robust to genetic perturbation, we reasoned that any dependencies that dedifferentiation has on specific cellular processes might be revealed by studying the relative timing of cellular events. Potential scenarios for the temporal organisation of progression through dedifferentiation might reflect a strict sequence of gene expression and cell level processes, or a less strict 'coming-into-being'. For a clear impression of the sequence of events, it is necessary to continuously follow individual cells through the dedifferentiation process.

We first addressed to what extent cell division was required for the gene expression changes of dedifferentiation, by scoring the first division time of individual cells. The first mitosis appeared considerably delayed compared to the onset of cell-cycle gene expression (*Figure 4—figure supplement 1A*). The median time of mitotic onset was 17.9 hr after dedifferentiation onset, contrasting the 5–6 hr at which cell cycle gene expression reached a plateau (*Figure 2—figure supplement 9*). More striking is that the division time corresponds to the stage at which the transcriptional changes of dedifferentiation are essentially complete (*Figure 1C*). These data suggest a considerable amount of post-transcriptional information processing is required before the first division can be activated, and indicates that division is not required for the majority of the transcript changes of dedifferentiation. After the first mitotic division, the overall duration of the second cell cycle (median 7.4 hr) was more similar to the undifferentiated cycle time (6.1 hr) (*Figure 4—figure supplement 1B*), suggestive of a near-complete return to the undifferentiated state, more in line with the gene expression time course data.

To what extent does the onset of division require the gene expression dynamics of dedifferentiation to unfold? To address this, we compared the onset of gene expression for the undifferentiated state with the timing of the first cell division (*Figure 4A*). As a marker of progression to the undifferentiated state, we used the *act8*-mNeonGreen reporter cells, additionally expressing mCherry-PCNA (*Miermont et al., 2019*) to facilitate monitoring of the cell cycle.

Dedifferentiating cells displayed considerable heterogeneity in both division time and *act8* induction profile (*Figure 4B* and *Figure 4—figure supplement 1A,C,D*). The bulk behaviour of *act8* induction indicated dividing cells induced expression more rapidly than non-dividing cells (*Figure 4B*) and in line with this, the overwhelming majority (*Figure 4C*) of dividing cells up-regulated *act8* before the first division. To further characterise the relationship between cell division and gene expression, we determined the time at which *act8* induction began and the subsequent rate of

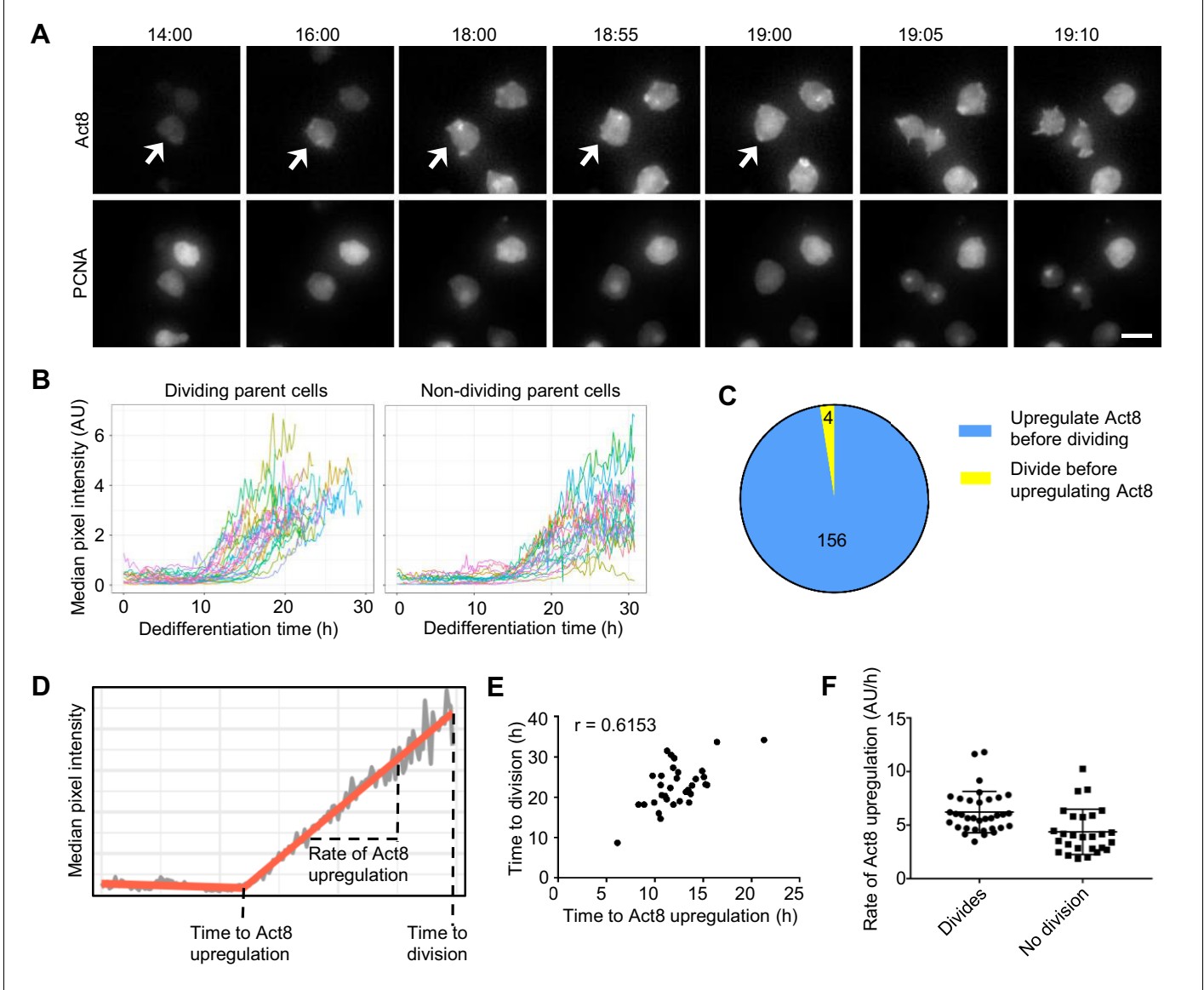

**Figure 4.** Single cell analysis of the coupling between events of dedifferentiation. (**A**) Example time lapse showing dedifferentiating amoebae expressing Act8 and PCNA reporters. Arrow indicates dividing cell. Scale bar = 10 μm. Time is hr:min. (**B**) Act8 reporter expression is induced earlier in dividing cells. Data from one representative experiment (354 cells over four experiments). (**C**) Proportion of cells that divided before or after onset of Act8 reporter expression, cell numbers indicated. (**D**) Schematic (based on real data) showing parameters extracted from Act8 expression traces and their relationship to the first cell division during dedifferentiation. (**E**) Correlation between time of Act8 expression onset and first division. Data from one experiment (36 divisions), representative of four independent experiments (160 divisions total, mean r = 0.5877). (**F**) Rate of increase of Act8 expression is higher in dividing than non-dividing cells. Data shown (36 dividing cells, 27 non-dividing) representative of four independent experiments (354 cells total). Mann-Whitney p value = 0.0002. Mean and SD are shown.

The online version of this article includes the following source data and figure supplement(s) for figure 4:

**Figure supplement 1.** Timing of gene expression changes and cell division during dedifferentiation in single cells.

**Figure supplement 1—source data 1.** Act8 reporter intensity tracks, cell fate data, Act8 reporter induction measurements, and cell cycle duration measurements.

increase in *act8* reporter expression for each cell (***Figure 4D***). The timing of the first mitosis was strongly correlated to the time of onset of *act8* induction (***Figure 4E***). The rate of act8 induction did not show a strong correlation with timing of the first mitosis across replicate experiments (r = −0.229, mean of four experiments), however the rate of *act8* induction was higher in dividing

cells than in those that did not divide (*Figure 4F*). Overall, these data suggest a strong degree of temporal coupling between the first mitosis and the onset of gene expression characteristic of the undifferentiated state, however the ordering of events is not absolute and the tendency of cells to carry out both processes either slow or fast is probably indicative of an overarching feature of cell state dictating dedifferentiation rate, rather than a strict sequence of events.

Another developmentally regulated property of cells is motility (*Varnum et al., 1986*). As cells become aggregation competent, they initially increase their motility, until aggregation, when their motile behaviour becomes more suited to migration within a 3D tissue, and so cells show reduced motility on surfaces. This tendency reverses during dedifferentiation, with cells increasing their speed over the first few hours, before slowing transiently, then becoming more motile again as dedifferentiation proceeds (*Figure 5A*). Dedifferentiating cells showed considerable heterogeneity in their motility, with speed profiles falling into two clusters (*Figure 5B* and *Figure 5—figure supplement 1A,B*). The majority (around 70%) of cells showed very little motility, whereas a small subpopulation showed a strong increase in migration, in line with the population average behaviour. Persistence was similar between the two clusters (*Figure 5—figure supplement 1C*). Monitoring the subsequent first mitoses of these cells revealed the motile population showed little tendency to divide during the 25 hr movie (*Figure 5C*). In contrast, most slow moving cells divided by the end of image capture. These observations of heterogeneity in the behaviour of cells reveal a slow moving dividing population and more rapid moving population that has deferred cell division. This might be consistent with some kind of bet-hedging response, with cells effectively speculating on staying put and proliferating, versus spreading in search of new habitats, reminiscent of heterogeneous motile behaviour in *Bacillus* and other bacteria (*Henrichsen, 1972*).

## Convergent gene expression trajectories during dedifferentiation

The heterogeneous cellular phenotypes during dedifferentiation suggested different cells might be using different gene expression trajectories. One possibility is that fast moving non-dividers and slow moving dividers may have their origins in the different starting fates of cells in the *Dictyostelium* aggregates, where 20% of cells become stalk cells, and 80% become spore.

To test this possibility, we imaged cell motility and division in a reporter cell line for the stalk fate. The *cryS* gene is strongly expressed in the prestalk lineage (*Antolović et al., 2019*) and we used a CryS-mNeonGreen knock-in reporter to follow the behaviour of the prestalk population during dedifferentiation. The cell line also expressed H2B-mCherry, which facilitates cell tracking. The starting expression level of CryS during dedifferentiation was not related to whether or not a cell divided during the 28 hr of image capture (*Figure 6A*), and the division time was uncorrelated to CryS level (*Figure 6B*). Finally, the motility of cells was not clearly linked to their CryS expression level (*Figure 6C*), with no significant differences between the cell speed distributions of the 20% cells with highest CryS expression and the remainder of the population (KS test: p=0.46 and 0.22 for two replicates). We conclude that the cell heterogeneity in motility and division time occurring during dedifferentiation arises independently of starting cell fate.

To further explore the potential for different dedifferentiation trajectories, we carried out single cell transcriptomics on the first 6 hr of dedifferentiation. Cells were collected from dedifferentiation cultures that had been set up at intervals, with cells then pooled into a single sample for cell capture. An overview of the data, showing the first two principal components, is shown in *Figure 6D* (see *Figure 6—figure supplement 1* for the replicate). The data show two major clusters, with the right cluster enriched for genes induced in the multicellular stage of development. Conversely, the left cluster of cells (negative PC1 values) shows a clear gradient of expression of genes repressed during development, such as ribosomal components. Comparing these data to the population transcriptomics data presented earlier implies PC1 reflects developmental time, with positive values corresponding to cells early in dedifferentiation, and negative values corresponding to cells closer to the undifferentiated state (*Figure 6E*). There was no clear evidence in this PCA space for multiple trajectories. Overlaying stalk and spore marker expression onto the PC1-PC2 space suggested the two fates occupied slightly different regions of the space, however, cells from the two fates were essentially on a similar path (*Figure 6F*). To distinguish more clearly between the starting fates required analysis of the higher order component, PC3.

Stalk and spore fates showed clear separation in PC1-PC3 space (*Figure 6G*). The set of prespore markers highlights different cells to the prestalk markers. Using PC1 as a proxy for developmental

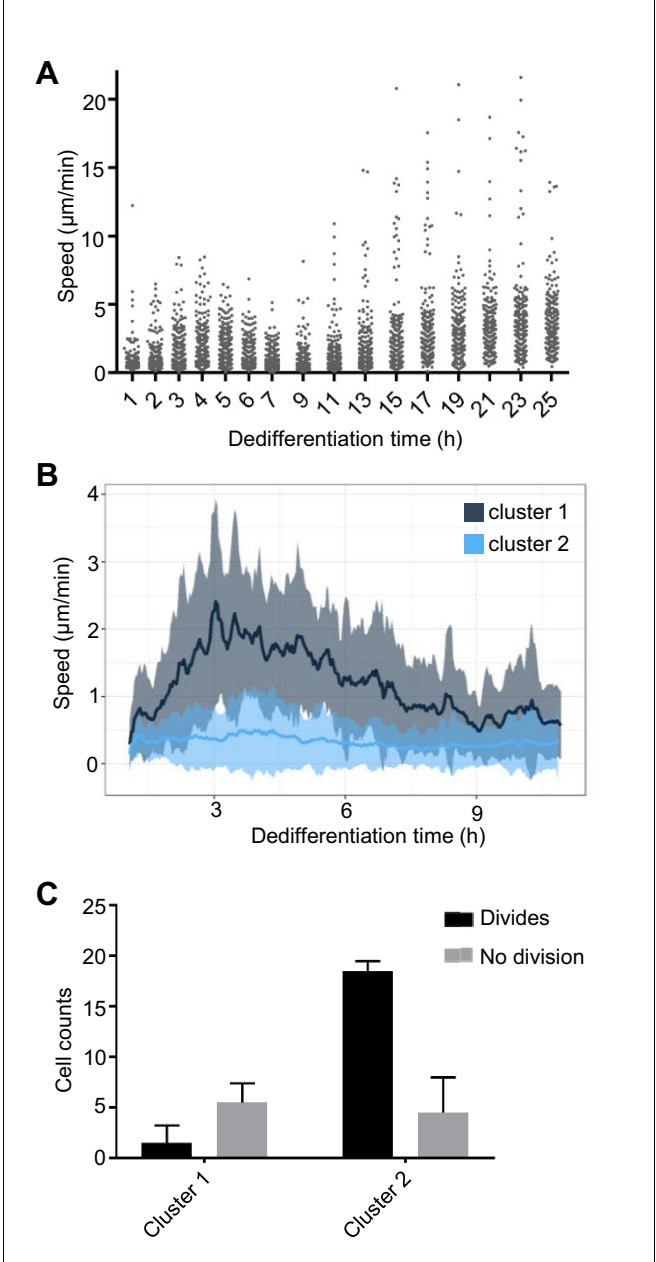

**Figure 5.** Coupling between cell motility and rate of dedifferentiation. (A) Regulation of cell motility during dedifferentiation. Cell speed was measured over 20 min windows, with image capture at 30 s intervals. 244–250 cells were captured for each time point, pooled from four replicates. (B) Distinct cell motility behaviours of cells during dedifferentiation. Two distinct clusters of cell speed profiles were identified. Speed is shown as a rolling average using a 10 min window. Line shows mean speed. Shaded area shows SD. Tracking used the same data as A, but cells were tracked continuously for the period shown rather than at intervals (120 tracks). (C) Slower cells are more likely to divide. Cells in the fast and slow moving clusters were scored for division or no division. The less motile cluster showed a greater tendency to divide during the period of image capture (25 hr). four independent imaging experiments, 30 cells per experiment. Mean and SD are shown. $\chi^2$ p<0.0001.

The online version of this article includes the following source data and figure supplement(s) for figure 5:

**Source data 1.** Cell speed, persistence and division data for motility experiments.
**Figure supplement 1.** Heterogeneity in cell motility during dedifferentiation.

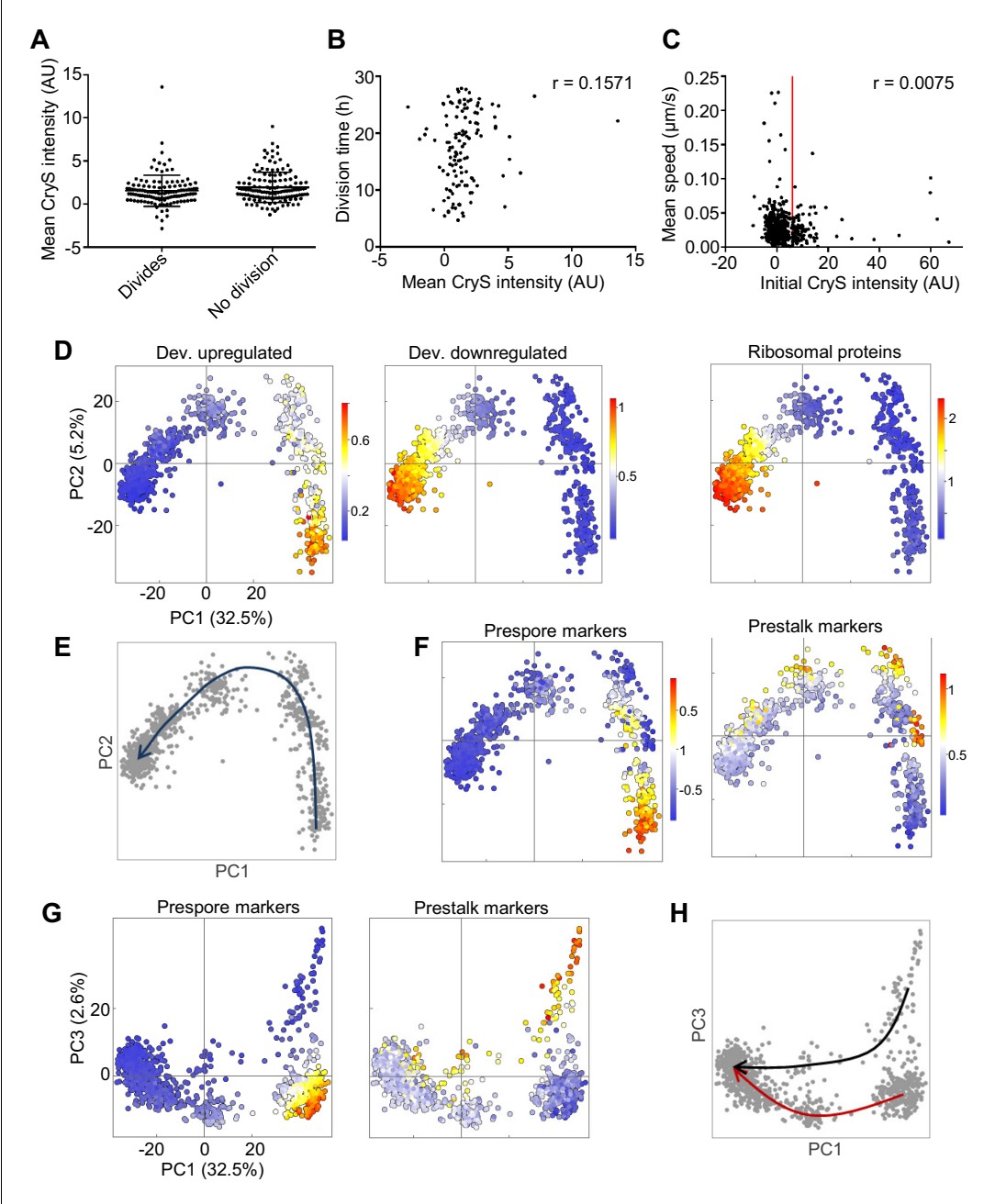

**Figure 6.** Rapid convergence of dedifferentiation trajectories. (**A**) The expression of the prestalk reporter (CryS-mNeonGreen) does not predict division probability during dedifferentiation. Reporter intensity at the beginning of dedifferentiation shows no significant difference between dividing and non-dividing cells (KS test p=0.134). Plots show mean and SD. One of two replicates shown (261 cells). (**B**) Initial fate and division time are not related. Relationship between initial CryS intensity and division time during dedifferentiation. One of two replicates shown (n = 124 divisions, r = 0.0864 mean of replicates). (**C**) Initial fate and motility are not correlated. Relationship between initial CryS expression and speed during the first 4 hr of dedifferentiation (442 cells). Vertical line indicates 80th percentile of CryS:mNeonGreen intensity. Mean r for two replicates = 0.0481. (**D**) Expression of different gene sets during dedifferentiation in 925 single cells (experimental replicate shown in *Figure 6—figure supplement 1*). PCA of scRNAseq overlaid with expression of a set of 303 developmentally induced genes (left panel), 276 genes turned off during development (centre panel) with 81 ribosome protein genes also shown (right panel). Each dot represents a cell. Cells were pooled from samples taken each hour during dedifferentiation (0–6 hr). (**E**) Schematic of the inferred path of cells during dedifferentiation. (**F**) Cell type specific gene expression during dedifferentiation. The same PCA plots as in D, but overlaid with the expression of sets of 42 prespore or 48 prestalk genes. (**G**) Cell type specific expression is more clearly delineated by PC3. The same expression data as in F plotted in PC1/PC3 space. (**H**) Convergence of cell type specific gene expression trajectories during dedifferentiation. Same plot as G, highlighted to show the inferred trajectories of cells with different starting fates. PCA colour scale indicates mean of log10 counts.

*Figure 6 continued on next page*

*Figure 6 continued*

The online version of this article includes the following source data and figure supplement(s) for figure 6:

**Source data 1.** CryS reporter intensity, cell division and cell speed data.

**Figure supplement 1.** Convergence of dedifferentiation trajectories.

time, prespore markers rapidly switch off, whilst the prestalk expression persists in cells with negative PC1 values. These observations are reminiscent of the behaviour of the different cell types during forward development, in which the prestalk cells appear less responsive (*Antolović et al., 2019*). The differing behaviours of prestalk and prespore cells during dedifferentiation do not interfere with the convergence of cell states (*Figure 6H*). By corroboration with Northern blot data of the *csaA* gene (*Figure 6—figure supplement 1D,E*), we find that the beginning of trajectory merging is around 3 hr after dedifferentiation onset. The later time points (3–6 hr) are relatively bunched, whereas the earlier time points are scattered. This implies, as suggested earlier, that most transcriptional changes occur early during dedifferentiation. Notably, the timescale of convergence is similar to that measured for the cell fate separation process during forward developmental progression (*Antolović et al., 2019*), implying no strong resistance to cell state change that is specific to dedifferentiation.

## Discussion

When compared to mammalian dedifferentiation contexts, the dedifferentiation response of *Dictyostelium* cells is remarkable in its speed and reliability, with most cells reversing their development in around 24 hr, whilst retaining their ability to generate a full complement of cell fates upon re-induction of development. In this study, we have surveyed the cell and molecular level dynamics associated with this efficient dedifferentiation response. The majority of gene expression changes constitute an apparently straightforward reversal of those occurring during forward development. The forward and reverse trajectories are not strict mirror image processes, with differences in the relative timing of events, and key transcriptional differences, notably a strong induction of ribosomal biogenesis components as cells return to the undifferentiated state. Cells from both fates appear to converge on the same gene expression trajectory rapidly during dedifferentiation, with no clear difference in the dedifferentiation phenotypes of spore and stalk-directed cells. The gene expression programme of dedifferentiation is robust to mutation of genes induced early in the programme, with relatively normal gene expression in mutants otherwise displaying strong proliferation defects.

The efficiency and apparent robustness may be related aspects of the dedifferentiation response. Dedifferentiation is induced experimentally by disaggregation of cells, followed by a nutritional stimulus. Disaggregation means a loss of stimuli related to cell contacts and dilution of other signals, such as cAMP. The nutritional stimulus, whether the peptone-based medium or bacteria (the more natural food source), is a complex mix of food biomolecules, likely to enter the cellular metabolic network via many pathways. Given the obvious complexity of the overall stimulus to dedifferentiate, it may be unlikely that the cell, regardless of its fate tendency, has any real choice but to efficiently obey this overwhelming sensory input, and that perturbation of any single molecule will be insufficient to substantially arrest the process. Along these lines, although we observed that different events during differentiation (gene expression and mitosis) show a distinct temporal sequence, at the single cell level the temporal ordering can be reversed. This implies that cell programmes can unfold in different ways to the same stimulus, potentially contributing to robustness. This lack of a rigid dependency between temporal phases of dedifferentiation may also explain why mutants may have some impairment in individual aspects of reversing development, whilst leaving the majority of dedifferentiation responses unperturbed (*Finney et al., 1983*; *Katoh et al., 2004*). This is exemplified by the phenotype of the *dhkA* mutant (*Katoh et al., 2004*). These cells have a delayed increase in cell number during dedifferentiation. The nature of this defect is unclear – is the effect on cytokinesis, or cell viability during suspension culture? Based upon the strong induction of the gene during development, its rapid repression during dedifferentiation, and the strong developmental phenotype of the mutant (*Wang et al., 1996*), a possible scenario is one in which the perturbed cell state during development feeds forward into an effect during dedifferentiation. Despite this phenotype,

as with our growth-defective mutants, other features of the *dhkA* mutant dedifferentiation response are normal. The *dhkA* mutants undergo erasure, replicate their DNA with normal timing and, based on plating efficiency of finger-stage cells, show wild-type levels of clonal recovery following dedifferentiation. Overall, these observations suggest the *dhkA* phenotype aligns well with the mutants described in our study (*forG* and *rasS*) that dedifferentiate fairly effectively despite an underlying cell health problem.

A recurring feature of the IPSC literature is the detection of developmental intermediates along reprogramming trajectories. These conclusions are usually based upon the expression of a few developmental markers. In *Dictyostelium,* we also identified the re-expression of markers from an early developmental time, in line with the mammalian literature, however the evidence was anecdotal, based on well-known genes, and did not stand up strongly to a more formal analysis. The idea that cells enter a pre-visited attractor state during reversal of development is an appealing idea if one wishes to understand how cells can navigate a path back to a progenitor- or fail to do so because they get trapped. The apparent lack of such a coherent developmental intermediate during dedifferentiation in *Dictyostelium* may to a certain extent underlie the speed and efficiency of developmental reversal.

The differences between dedifferentiation in *Dictyostelium* and during IPSC generation appear to be systemic, and it seems unlikely at the present time that we could use knowledge from the former to improve the latter. The efficiency of dedifferentiation in *Dictyostelium* may ultimately come down to the likelihood that it is a physiological response, whereas the generation of IPSCs is clearly not. In the migratory slug phase of development many cells fall out of the rear of the slug onto the substrate, where they are then able to re-enter the feeding part of the lifecycle, if bacteria are present, and then fully re-differentiate once the supply of bacteria is exhausted (*Kuzdzal-Fick et al., 2007*). This is in effect a dispersal strategy, bet-hedging against the potential for a proliferative disadvantage occurring within the dormant spore state. Meaningful parallels between the dedifferentiation responses of *Dictyostelium* and mammalian cells seem more likely in situations where mammalian cells dedifferentiate as part of the normal course of events, such as during responses to tissue damage and metabolic stress.

## Materials and methods

### Cell lines

*Dictyostelium* cells were grown in HL5 (Formedium). Wild type AX2 cells were used unless otherwise stated. Cell lines were authenticated using PCR and Southern blotting.

Cells maintained in HL5 were kept under selection against bacterial contamination. Cultures were not allowed to enter stationary phase and cells were not used beyond 10 days of culture. For experiments with cells defective in growth in HL5 (*forG-*, *rasS-* and *DDB_G0280067-* mutants), both mutant and wild-type cells were grown on *Klebsiella* on SM agar and harvested prior to clearance of the bacterial lawn, with cells washed free from bacteria before further processing.

For development, log-phase cells were washed in KK2 (20 mM $KPO_4$ pH 6.0) and developed on Whatman #50 filter paper at a density of $2.6 \times 10^6$ cells cm$^{-2}$ in a humidified chamber. For dedifferentiation, 14 hr developed cells were washed from the filter paper in KK2 + 20 mM EDTA, then disaggregated by repeated passing through a 20G needle. Cells were dedifferentiated either in HL5 suspension culture (at $2 \times 10^6$ cells ml$^{-1}$) or in culture dishes (at $4 \times 10^6$ cells cm$^{-2}$) containing live *Klebsiella* in KK2 (OD600 = 2). Buffer-only treatments were carried out in KK2 in culture dishes.

Mutants in candidate dedifferentiation regulators were made by homologous recombination. The details of the targeting vectors used are described in the Appendix. Resistance to Blasticidin S was used as the basis for selection of recombinants (*Faix et al., 2004*). Mutants in *rasG* (*Veltman et al., 2016*), *pakE* (*Sawai et al., 2007*), *gefS* and *krsB* (*Williams et al., 2019*) and *rasS* (*Chubb et al., 2000*) were described previously.

### RNAseq

For population RNAseq, RNA was prepared from cell pellets as described (*Chubb et al., 2000*). Processing of RNA samples for sequencing, and read mapping was carried out as described in the Appendix. Read counts were normalised using the size factor calculated with DESeq2 package

(*Love et al., 2014*). Mean values of the two replicates were used for analysis, unless otherwise stated. PCA was performed in R, using only genes with mean read counts >10 (10143 genes for PCA in *Figure 1C* and 10063 for *Figure 2A*). For hierarchical clustering of the genes with the greatest contribution to PC1 (*Figure 1D*), we ranked the genes by their loading, then used the top-ranked genes whose total contribution gives rise to 10% of the component's variance (total 580 genes). For more general characterisation of samples in principal component space, we used the top-ranked genes whose total contribution gives rise to 25% of the component's variance (1518 and 1220 genes for PC1 and PC2, respectively). For estimating overlap of forward and reverse trajectories, we considered the portion of the trajectories with the biggest change in PC1 values. For development, we used the 2–6 hr time points and for dedifferentiation, we used the 0.5 hr to 4 and 5 hr in liquid medium and bacteria respectively. Two-way hierarchical clustering of genes changing during dedifferentiation on bacteria was carried out in MATLAB. Overall, we defined the genes changing during dedifferentiation as those satisfying the following conditions: the normalised read count was >100 in at least one time point and the $|\log_2 FC|$ between 0 hr and at least one other time point was >1. This provided 6574 genes for liquid medium and 7174 genes for bacterial culture treatments. Gene Ontology enrichment analysis used PANTHER Classification System version 14.1.

For single cell RNAseq, cells were dedifferentiated in HL5 suspension. The start time of development in the samples was staggered such that separate cultures at 0, 1, 2, 3, 4, 5 and 6 hr of dedifferentiation could be simultaneously collected and resuspended in ice-cold KK2. Detailed information on single cell processing using the Chromium system (10x Genomics), sequencing and data analysis is described in the Appendix. For interpreting PCA, cell-type specific genes were selected from published population transcriptomic data (*Parikh et al., 2010*) with $|log_2 FC|>1$, $FDR<0.1$ and an expression level of >100 normalised molecular counts in at least one cell. This gave 42 prespore markers and 48 prestalk markers. Sets of genes being up- or down-regulated during the mound stage of development are taken from *Antolović et al. (2019)*.

Sequencing data have been deposited at GEO with accession number GSE144892.

## Imaging

Imaging experiments were carried out using cells dedifferentiating in bacteria. For monitoring onset of *act8* expression in relation to cell division, we used Act8-mNeonGreen knock-in cells (*Tunnacliffe et al., 2018*), transformed with an mCherry-PCNA expression plasmid (*Miermont et al., 2019*). Cells were plated at a density of $2.4 \times 10^4$ cells cm$^{-2}$ on chambered coverglass (Nunc) and imaged on an inverted microscope optimised for fast sensitive imaging (*Muramoto and Chubb, 2008*), with capture of 30 slice z stacks with a 0.5 µm step-size for at least 25 hr, using a dual GFP/mCherry filter set (Chroma 59022), with 50 ms exposure per slice, per channel.

For assaying cell motility during dedifferentiation, cells were imaged using phase contrast on a Zeiss Observer Z1 inverted microscope with automated Prior stage and Hamamatsu Orca-Flash4.0 camera using a 10x Plan-Neofluar Ph1 objective, with two frames captured per minute for 25 hr.

To compare cell fate to dedifferentiation features, we used a mNeonGreen knock-in reporter for the early prestalk marker CryS. The reporter was targeted into AX3 cells previously modified to express the fluorescent nuclear marker H2B-mCherry (*Corrigan and Chubb, 2014*). Cells were imaged on a custom built inverted wide field microscope, equipped with Prime 95B CMOS camera (Photometrics), 10x UplanFL N objective (Olympus) and 470 nm and 572 nm LED light sources (Cairn Research). 3D stacks spanning 20 µm over five slices were captured every 2 min for 28 hr over a 3 × 3 grid of adjacent fields of view.

Comprehensive protocols for image analysis are documented in the Appendix.

## Gene expression and signalling assays

Dedifferentiation of mutants in Act8-mNeonGreen cells was assayed by monitoring mNeonGreen fluorescence at 0 hr and 24 hr of dedifferentiation by flow cytometry (BD Biosciences LSRII). Two independent clones of each mutant were both tested in two separate experiments, with >50000 cells measured for each strain at each timepoint. FlowJo v10 software (FlowJo, LLC) was used for data analysis.

For assaying other mutants, we used RNAseq (see above) or more widely, Northern blotting of RNA extracted from cells during dedifferentiation in HL5 suspension. For analysis of AMPK and

mTORC1 signalling, protein extracted from cell pellets of dedifferentiating cells was blotted using anti-pAMPKα (Thr 172, clone 40H9 rabbit mAb, CST#2535) and anti-p4E-BP1 (Thr 37/46, rabbit Ab, CST#9459). For a standard, we blotted parallel-loaded extracts using an antibody against the C-terminus of histone H3 (Abcam #ab1791). Further details regarding blotting can be found in the Appendix.

## Macropinocytosis

Fluid uptake measurements were adapted from a standard protocol (*Wilkins et al., 2000*). Cells were dedifferentiated in HL5 suspension culture at $2 \times 10^6$ cells ml$^{-1}$. At the indicated times, aliquots were removed and adjusted to $3 \times 10^6$ cells ml$^{-1}$ in 3 ml. In parallel, undifferentiated cells were taken from a mid-log suspension culture and adjusted to the same density. For both culture types, 2 mg ml$^{-1}$ TRITC-dextran (65–85 kDa, Sigma) was added, with a further 3 ml of each culture retained as an unlabelled control. After 1 hr labelling, fluorescence was quenched with Trypan Blue, while the unlabelled control was quenched immediately after addition of dextran. Cells were washed and resuspended at the same density in ice cold KK2. Internalised fluorescence was measured using a Fluoromax+ spectrofluorometer (Horiba). Fluorescence measurements of unlabelled cells were subtracted from measurements of labelled samples, and fluorescence uptake during dedifferentiation expressed as a percentage of uptake measured in undifferentiated cells.

## Acknowledgements

This work was supported by Wellcome Trust Senior Fellowship 202867/Z/16/Z to JRC and Medical Research Council (MRC) funding (MC_U12266B) to the MRC LMCB University Unit at UCL. Imaging was carried out at the MRC LMCB Light Microscopy Facility. Sequencing was carried out at the Barts and the London Genome Centre.

## Additional information

### Funding

| Funder | Grant reference number | Author |
| --- | --- | --- |
| Wellcome | 202867/Z/16/Z | Jonathan R Chubb |
| Medical Research Council | MC_U12266B | Jonathan R Chubb |

The funders had no role in study design, data collection and interpretation, or the decision to submit the work for publication.

### Author contributions

John ME Nichols, Conceptualization, Resources, Data curation, Validation, Investigation, Methodology, Writing - original draft, Project administration, Writing - review and editing; Vlatka Antolović, Conceptualization, Resources, Data curation, Software, Formal analysis, Validation, Investigation, Visualization, Writing - original draft, Writing - review and editing; Jacob D Reich, Software, Formal analysis, Investigation, Writing - review and editing; Sophie Brameyer, Investigation; Peggy Paschke, Resources; Jonathan R Chubb, Conceptualization, Resources, Supervision, Funding acquisition, Validation, Investigation, Methodology, Writing - original draft, Project administration, Writing - review and editing

### Author ORCIDs

John ME Nichols (iD) https://orcid.org/0000-0003-2061-1485
Vlatka Antolović (iD) https://orcid.org/0000-0002-1943-1850
Sophie Brameyer (iD) http://orcid.org/0000-0002-6779-2343
Jonathan R Chubb (iD) https://orcid.org/0000-0001-6898-9765

### Decision letter and Author response

Decision letter https://doi.org/10.7554/eLife.55435.sa1

Author response https://doi.org/10.7554/eLife.55435.sa2

## Additional files

### Supplementary files

- Supplementary file 1. Key Resources Table.

- Transparent reporting form

### Data availability

Sequencing data have been deposited to GEO under the accession number GSE144892.

The following dataset was generated:

| Author(s) | Year | Dataset title | Dataset URL | Database and Identifier |
|---|---|---|---|---|
| Nichols JME, Antolovic V, Reich J, Brameyer S, Paschke P, Chubb JR | 2020 | Cell and molecular transitions during efficient dedifferentiation | https://www.ncbi.nlm.nih.gov/geo/query/acc.cgi?acc=GSE144892 | NCBI Gene Expression Omnibus, GSE144892 |

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

## Appendix

### Generation of mutant cell lines

Knockout mutants were made by homologous recombination. Unless otherwise stated, we PCR amplified two arms of homology surrounding or within each gene from genomic DNA, and inserted a blasticidin resistance cassette (bsr) between the two arms: *gefAA* (+297 to +999, +1334 to +2158), *DDB_G0272434* (+556 to +1343, +1502 to +2262), *DDB_G0280067* (+87 to +792, +815 to +1435), *DDB_G0275621* (+54 to +755, +807 to +1520), *DDB_G0279851* (−692 to +33, +65 to +591), *DDB_G0283057* (+72 to +897, +981 to +1742), *DDB_G0288203* (−1 to +579, +681 to +1424), *DDB_G0276549* (+85 to +868, +914 to +1737), *DDB_G0268696* (+150 to +926, +1004 to +1716), *tagA* (+462 to +1269, +1666 to +2500), *DDB_G0274177* (+396 to +1136, +1581 to +2267), *DDB_G0289907* (+372 to +1128, +1622 to +2424), *DDB_G0269040* (+1127 to +1871, +2121 to +3001), *DDB_G0272364* (+865 to +1689, +2164 to +3016), *gtaN* (+365 to +1267, +1896 to +2824), *xacB* (−1227 to −273, +4744 to +6116), *mybD* (+one to +835, +1003 to +1788), *DDB_GO269374* (+96 to +674, +789 to +1495), *nfyA* (−103 to +626, +649 to +1358), *bzpS* (−647 to +29, +934 to +1733), *DDB_G0272386* (−966 to +30, +86 to +1025), *bzpI* (+one to +1502 (resistance cassette inserted internal EcoRI site in one fragment)), *fslN* (−830 to +107, +160 to +823), *gbpD* (−469 to +29, +76 to +1018), *jcdA* (+one to +509, +517 to +1216), *nfaA* (+514 to +1262, +1348 to +2054), *ptpB* (−624 to +25, +409 to +1344), *DDB_G0277531* (−303 to +1099 (internal EcoRI site used for resistance cassette)), *eriA* (−548 to +206, +226 to +1076), *sodC* (−45 to +579, +612 to +1198), *omt5* (−450 to +392, +444 to +977), *zacA* (+one to +740, +800 to +1546), *DDB_G0292302* (+one to +629, +645 to +1271), *DDB_G0293562* (655 to +29, +27 to +649), *sigB* (+59 to +811, +814 to +1601), *DDB_G0278193* (−9 to +707, +760 to +1486), *DDB_G0293078* (−555 to +197, +212 to +923), *DDB_G0270480* (+274 to +920, +976 to +1692). The following genes were disrupted by insertion of an mNeonGreen-bsr cassette: *ctnB* (−977 to +26, +977 to 1750) and *DDB_G0270436* (−877 to +48, +2043 to +2875). The *forG* targeting vector was generated by insertion of the hygromycin cassette (derived from pDM1081) between −745 to −72 and +70 to +845 of the genomic sequence.

The combined fragment comprising resistance cassette flanked by arms of homology was released by restriction digest for transformation into *Dictyostelium* AX2 cells. Transformants were selected in the presence of 10 µg ml$^{-1}$ blasticidin S, or 35 µg ml$^{-1}$ hygromycin B, and clonal populations screened by PCR. Gene disruption was confirmed by Southern blotting, with exception of *xacB* where deletion was confirmed by PCR across the whole locus.

We did not recover simple insertional mutants in the following 17 genes: *DDB_G0284069*, *abcF3*, *DDB_G0279205*, *DDB_G0272450*, *DDB_G0295757*, *gefP*, *DDB_G0278179*, *hbx10*, *mybH*, *gtaE*, *DDB_G0268506*, *DDB_G0274691*, *DDB_G0293064*, *rio1*, *DDB_G0290815*, *DDB_G0270038*, *DDB_G0269344*.

### Population RNAseq

RNA samples were assessed for quantity and integrity using the NanoDrop 8000 spectrophotometer V2.0 (ThermoScientific, USA) and Agilent 2100 Bioanalyser (Agilent Technologies, Waldbronn, Germany), respectively. 100 ng of total RNA from each sample was used to prepare mRNA libraries using the NEBNext mRNA isolation kit in conjunction with the NEBNext Ultra (Ultra II for transcription factor mutants) Directional RNA Library preparation kit (New England Biolabs, Massachusetts, USA). Fragmentation of isolated mRNA prior to first strand cDNA synthesis was carried out using incubation conditions recommended by the manufacturer for an insert size of 300 bp (94℃ for 10 min). 13 cycles of PCR were performed for final library amplification. Resulting libraries were quantified using the Qubit 2.0 spectrophotometer (Life Technologies, California, USA) and average fragment size assessed using the Agilent 2200 Tapestation with D1000 screentape (Agilent Technologies, Waldbronn, Germany). A final sequencing pool was created using equimolar quantities of each sample with compatible indexes. 75 bp paired-end reads were generated for each library using the

Illumina NextSeq500 in conjunction with a Mid-output 150-cycle kit or NextSeq500 v2 High-output 150-cycle kit (Illumina Inc, Cambridge, UK).

Read quality was checked using FASTQC. Reads were mapped to the *Dictyostelium* genome (version obtained from Gareth Bloomfield, masking the duplication on chromosome 2) using Tophat v2.0.9. For mapping, the library type parameter was set to 'fr-firststrand', as appropriate for the dUTP method used in the library preparation kit. Mapped reads were assigned to gene models using HtSeqCount v0.5.4p3. Here we specified that reads came from a directional first-strand enriched library by setting the stranded parameter as 'reverse', so that read pairs had to be mapped in the correct orientation in order to be assigned to a feature. For handling overlapping features, we used HtSeqCount in 'union' mode. This means that reads mapped wholly or in part to multiple features were treated as ambiguous and as a result were not assigned to a feature. To visualize the location of mapped reads we used IGV (v2.6.3).

## Single cell RNAseq

Cell suspensions were loaded to the 10X Chromium Single Cell A Chip (PN-1000009) using the Chromium 3' Library and Gel Bead Kit v2 (PN-120267) as described by the manufacturers (10X Genomics, California). 14 cycles of cDNA amplification were performed on the purified GEM-RT product, and cDNA was examined for quality using the Agilent 2200 Tapestation with the High-sensitivity D5000 screentape and reagents (Agilent Technologies, Waldbronn, Germany), and the Qubit 2.0 Fluorometer and Qubit dsDNA HS Assay Kit (Life Technologies, California, USA). 35 µL of cDNA was used to prepare the $10 \times 3'$RNA library and 12 and 11 cycles were used for sample index PCR of replicates 1 and 2 respectively. Final cleaned libraries were quantified using the Qubit 2.0 Fluorometer and Qubit dsDNA HS Assay Kit and average fragment size checked using the Agilent D1000 screentape and reagents. The final library was run on a NextSeq500 Mid-output 150-cycle kit with a 26[8]98 cycle configuration to generate 130 million read pairs in total.

## Analysis of scRNAseq data

Alignment, barcode counting, UMI counting and filtering was performed by Cell Ranger v2.2.0 using default parameters. A total of 967 and 2961 single cell libraries passed the filter, with a median of around 25000 and 18000 total molecular counts (UMIs), for replicate one and two, respectively.

We excluded outliers with high sequencing depths (three interquartile ranges above the third quartile), cells missing a contiguous part of the transcriptome and cells with less than 2000 mapped genes. A total of 925 and 2415 cells, from replicate one and two, respectively, were used for further analysis.

Molecular counts of cells within each replica were normalised using the size factor calculated with DESeq2 package (*Love et al., 2014*). PCA analysis was performed in R (only genes with mean >1 were used, 2979 for replicate 1 and 2386 for replicate 2). The visualisation of gene expression analysis was done in Mathematica.

## Reagents for blotting

For probe templates for Northern blotting, we used fragments spanning coding sequence released by digestion from overexpression or targeting vectors: for *PCNA*, we used an EcoRI fragment from a GFP-PCNA expression vector (*Muramoto and Chubb, 2008*); for *rpl15*, we used a BamHI/NotI fragment from the *rpl15*-MS2 targeting vector (*Muramoto et al., 2012*); for *hspE,* we used +993 - +1891 of the coding sequence; for *sodC* we used a fragment cloned from genomic DNA spanning +452 to +1339; for H2Bv1 we used the entire coding sequence cloned from genomic DNA; for *csaA*, we used a BamHI/Not fragment from a *csaA* knock-in targeting vector (*Muramoto et al., 2012*). For size estimation of *PCNA* transcripts, we used

0.1 kb to 2 kb (Invitrogen, cat. #15623100) and 0.5 kb to 10 kb (Invitrogen, cat. #15623200) RNA ladders.

For Western blotting, cell pellets were lysed on ice in RIPA buffer containing protease (Complete Ultra, Roche) and phosphatase (PhosSTOP, Roche) inhibitors. Protein concentration was determined by BCA assay (Thermo) and equalised between samples by addition of RIPA buffer containing protease and phosphatase inhibitors. LDS sample buffer (1x final, Invitrogen) and β-mercaptoethanol (5% v/v final) were added and samples denatured by boiling for 5 min. 30 µg total protein per sample was used for SDS-PAGE in 1x MES SDS running buffer (Invitrogen). Proteins were transferred onto nitrocellulose (Protran 0.2 µm pore size, GE) at 35V for 1.25 hr in a Nupage transfer system (Invitrogen). Equal loading and transfer of total protein was assessed by Ponceau S (Sigma) staining of the membrane. Membranes were blocked with 5% (w/v) BSA in TBS-T, then incubated with primary antibody diluted in TBS-T with 5% (w/v) BSA overnight at 4°C. For detecting phospho-AMPK we used a 1/500 dilution of anti-pAMPK alpha (Thr 172, clone 40H9 rabbit mAb, CST#2535). For detecting the mTORC1 substrate p4E-BP1, we used a 1/500 dilution of anti-p4E-BP1 (Thr 37/46, rabbit Ab, CST#9459). After washing in TBS-T, membranes were incubated with anti-rabbit IgG HRP-linked secondary antibody (whole antibody from donkey, GE healthcare) in TBS-T + 5% (w/v) BSA. Secondary antibody dilutions were 1/10000 for pAMPK, 1/5000 for p4E-BP1. Blots were washed in TBS-T before chemiluminescent detection (Supersignal West Femto, Thermo).

## Image analysis

For analysis of *act8* induction, sum intensity projections of z-stacks had rolling ball (50 pixel) background correction applied using FIJI. Cells were manually tracked and mNeonGreen intensity measured on the rolling ball data using FIJI plugin Time Series Analyzer v3, with identification of divisions aided by the localisation pattern of PCNA. To extract the timing of *act8* up-regulation the R package 'segmented' was used to iteratively fit segmented linear regressions (with a breakpoint) to mNeonGreen intensity data. For cells where *act8* reached a post-induction plateau before division/end of track, a three segment regression with two breakpoints was used.

For measuring cell speed, cells were manually tracked in FIJI using the Manual Tracking plugin. To sample cell speed in the population, cells were tracked for 20 min windows at intervals throughout dedifferentiation. For long-term tracking of speed and division in individual cells, cells were tracked at a frame rate of 2 min per frame between 1 hr and 11 hr of dedifferentiation, then followed until 25 hr dedifferentiation to score division. Instantaneous speeds for each cell were averaged over a 10 min rolling window. Persistence was calculated for each cell as the ratio of Euclidean and accumulated distance over a sliding window of 30 min. Cell speed profiles were k-means clustered in R using the package 'cluster', with k = 2 selected for k-means clustering based on the silhouette method. PCA of speed profile clusters used the R package 'factoextra'.

For comparing cell fate and division time, the initial level of CryS and division time for each cell were determined manually using a custom ImageJ macro. The mean intensity in the CryS channel was determined within a 12 pixel radius of the nuclear position, followed by background subtraction, with the median signal of a nearby 30 pixel radius region, selected by eye, used as background. Each cell was manually tracked, recording the first division or marked as 'non-dividing'.

For comparing cell speed and CryS level, nuclei were automatically tracked, and speeds and CryS-neon fluorescence intensity measured, using custom Python scripts and particle tracking library Trackpy1 (code available at https://github.com/Sloth1427/10x_analysis; *Reich, 2020*; copy archived at https://github.com/elifesciences-publications/10x_analysis). CryS levels were calculated as the mean of a 20-pixel radius around the nuclear position, minus the background (the median pixel value across the entire image, excluding a 40-pixel radius around nuclei). The initial CryS level for each nucleus was calculated as the mean CryS level for the first five frames (10 min). Average speed was calculated for each complete cell track present during consecutive time windows during dedifferentiation.

Evaluating potential relationships between different features of gene expression and cell physiology used Pearson's correlation values.

