## [Decision Letter]

**Acceptance summary:**

This is a very nice study expanding on dedifferentiation in Dictyostelium; it will yield key insights into dedifferentiation in other systems.

**Decision letter after peer review:**

Thank you for submitting your article "Cell and molecular transitions during efficient dedifferentiation" for consideration by *eLife*. Your article has been reviewed by three peer reviewers, one of whom served as a guest Reviewing Editor, and the evaluation has been overseen by Didier Stainier as the Senior Editor. The following individuals involved in review of your submission have agreed to reveal their identity: Derrick Brazill (Reviewer #2); Pauline Schaap (Reviewer #3).

The reviewers have discussed the reviews with one another and the Reviewing Editor has drafted this decision to help you prepare a revised submission.

Summary:

The reviewers find this work interesting, but have requested several clarifications. Please address all of the points raised by the three reviewers.

Essential revisions:

1) Results subsections “Genome scale features of dedifferentiation” and “Staging gene expression during dedifferentiation” – since this seems to largely follow the Katoh paper, you should point out similarities and differences.

2) Subsection “Genetic robustness of the dedifferentiation programme” – the authors say they cannot find mutants that affect dedifferentiation, but Katoh et al. showed that the *dhkA* mutant strongly affects dedifferentiation, I'm very puzzled.

Reviewer #1:

This report examines gene expression in D. discoideum cells when the cells are starved to induce differentiation, and then transferred to nutrient-rich growth medium to induce dedifferentiation. There are strong similarities but puzzling differences between the first part of this report and a PNAS paper previously published by another lab, and the similarities and differences need to be clarified.

1) Results subsections “Genome scale features of dedifferentiation” and “Staging gene expression during dedifferentiation” – since this seems to largely follow the Katoh paper, you should point out similarities and differences.

2) Subsection “Genetic robustness of the dedifferentiation programme” – the authors say they cannot find mutants that affect dedifferentiation, but Katoh et al. showed that the *dhkA* mutant strongly affects dedifferentiation, I'm very puzzled…

Reviewer #2:

This manuscript addresses an interesting and important question regarding the process of dedifferentiation. Little is known about the process except in very artificial systems, so anything learned about it in a natural system has the potential to be very significant. To address this, the authors use Dictyostelium discoideum, which will halt and reverse its starvation-mediated developmental program if the cells in the mound are dissociated and provided food. With this system, they ask a basic question: Is dedifferentiation simply differentiation in reverse, or something different?

While this question has been examined in Dictyostelium, the authors do a much more thorough an in depth examination into the transcriptional changes associated with differentiation and dedifferentiation. It is the power of this approach that makes their finding that in general dedifferentiation correlates with a reversal of developmental gene expression valuable and worthy of publication.

Unfortunately, their attempts at identifying the genetic regulators of the process were unsuccessful. The genetic screen using act8-mNeonGreen was inspired, even though it did not work. However, the remainder of the focused didn't seem as logical. While the choice of protein types (signaling, transcription, cell division and growth) made sense, the chosen genes seemed random. There were much more rational choices within these groups. DhkA is a developmentally regulated signaling protein that is known to be involved in dedifferentiation. ChdA, ChdB and ChdC are supposed chromatin remodeling proteins that are master regulators for developmental gene expression. There are multiple G α proteins that are known to be involved in food sensing during growth. Almost all of the mutants of all of these gene are able to reach the mound stage of development, so can be examined in this dedifferentiation system. While I am not suggesting that the authors perform these experiments in order to have this manuscript published, I would like a clearer rationale for the mutants that they did choose and why they are better choices than others.

I would also be wary of claiming that the dedifferentiation programme is genetically robust for the same reasons. Even though the multitude of mutants they tested didn't show large effects on dedifferentiation, they left out some fairly obvious ones. It would be safer to say that dedifferentiation is more genetically robust that differentiation, since there are a lot more genes that disrupt differentiation. However, this probably has more to do with the fact that development and differentiation requires a higher degree of temporal coordination than dedifferentiation. For dedifferentiation, it makes sense that it doesn't matter whether regulated cAR1 expression occurs before or after adenylyl cyclase expression or G α 2 expression. However, during development, the relative timing of expression is crucial.

The cryS experiments were critical in demonstrating that cell fate was not responsible for the two different populations of dedifferentiated cells identified based on motility and division speed. I would ask that the control be mentioned showing proper CrS-mNeonGreen expression in the mounds before dissociation otherwise we don't know whether the system worked well-enough to identify actual prestalk cells. I'd also request a reanalysis of the data to see whether the high/low CryS-mNeonGreen expressing cells had the same ratios of high speed/low speed cells as the entire population. It may be that there is an enrichment of one type of cell based on CryS-mNeonGreeen expression. The data should already exist. It would just need to be analyzed differently.

Reviewer #3:

This work focusses on dedifferentiation, an overlooked but important process that is important for wound healing, stem cell replenishment and cancer progression.

The authors first use an RNAseq approach to compare forward development and dedifferentiation and find that while occurring at a slower pace, the reverse trajectory is the mirror image of the forward trajectory for most genes, but that this is less so for genes involved in protein degradation and ribosome biogenesis.

They investigate the hypothesis that dedifferentiation revisits earlier developmental stages, which is suggested by the behaviour of a relatively small number of developmental markers in both pluripotent stem cell induction and their current study. However, for a larger set of 402 genes involved in the aggregation process that precedes the chosen endpoint of forward development, the extent to which this holds true varies from 41% to 22% of genes, dependent on whether dedifferentiation is induced by exposure to bacterial food or to liquid nutrient medium. This suggest that also for animal cells this hypothesis needs to be revisited.

The authors use several approaches to show that the dedifferentiation process is robust and not easily perturbed by single gene deletions, such as those of transcription factors and signaling genes that are up-regulated at the onset of dedifferentiation, or identified from mutants that fail to up-regulate the act8 gene that is only expressed in undifferentiated cells.

A single cell RNAseq approach was used to determine the relationship between cell division and expression of cell cycle genes. This showed that cell division lagged 12 h behind and occurred when cells had almost completely dedifferentiated, which supported by additional experimentation indicated that cell division is not required for dedifferentiation.

Motility followed expression of motility genes more closely, but surprisingly only a fraction of the cells showed the expected increase in motility during dedifferentiation. This appeared unrelated to the prespore or prestalk cell fate that the cells had acquired. The latter also had no effect on the timing of cell division during dedifferentiation nor on the path taken by the two cell types to reach the dedifferentiated state.

These results contrast quite strikingly with forward development where cell fate is correlated with several parameters that vary in early stage cells, such as cell cycle phase and nutritional state.

All in all the study demonstrates interesting similarities and differences between forward and reverse differentiation in Dictyostelia, which can be expected to be true for other multicellular organisms as well. The experimentation and data analysis is of a very high standard with important conclusions being supported by complementary approaches.

---

## [Author Response]

Essential revisions:1) Results subsections “Genome scale features of dedifferentiation” and “Staging gene expression during dedifferentiation” – since this seems to largely follow the Katoh paper, you should point out similarities and differences.

We have incorporated several additional references to the Katoh paper in this section, and in the manuscript as a whole, to better contextualize our studies.

2) Subsection “Genetic robustness of the dedifferentiation programme” – the authors say they cannot find mutants that affect dedifferentiation, but Katoh et al. showed that the dhkA mutant strongly affects dedifferentiation, I'm very puzzled.

A closer analysis of the Katoh paper, together with further analysis of our own RNAseq data, provides an improved contextualization of the phenotype of the *dhkA* mutant:

1) All but one of the dedifferentiation features of the *dhkA* mutant showed wild-type behaviour.

2) The one phenotype apparent in the *dhkA* mutants was a delay in the increase in cell counts during dedifferentiation, although it was not clear from the Katoh data whether this apparent growth defect was specific to dedifferentiation, or a more general growth defect caused by loss of *dhkA*. In several papers from the Shaulsky/Loomis labs, the *dhkA* mutant displays a very strong developmental defect (e.g. EMBO J15: 3890-8). One possibility is that the delay in increase in cell number during dedifferentiation is a knock-on effect of the defective cell state during development.

3) Our RNAseq data shows that *dhkA* is strongly induced during development, but strongly and rapidly repressed during dedifferentiation, under all culture conditions. This implies a developmental role rather than a dedifferentiation role for the protein, again supporting the view that the *dhkA* phenotype during dedifferentiation is non-specific. Overall, we feel the *dhkA* phenotype fits well with the mutants described in our study (*forG* and *rasS)* that dedifferentiate fairly effectively despite an underlying cell health problem.

We have incorporated this reasoning into an extended paragraph in the Discussion. The *dhkA* expression profiles during development and dedifferentiation are now included as a new Figure 3—figure supplement 1A. We have also added a section in the Introduction to introduce the *dhkA* mutant.

Reviewer #1:This report examines gene expression in D. discoideum cells when the cells are starved to induce differentiation, and then transferred to nutrient-rich growth medium to induce dedifferentiation. There are strong similarities but puzzling differences between the first part of this report and a PNAS paper previously published by another lab, and the similarities and differences need to be clarified.1) Results subsections “Genome scale features of dedifferentiation” and “Staging gene expression during dedifferentiation” – since this seems to largely follow the Katoh paper, you should point out similarities and differences.2) Subsection “Genetic robustness of the dedifferentiation programme” – the authors say they cannot find mutants that affect dedifferentiation, but Katoh et al. showed that the dhkA mutant strongly affects dedifferentiation, I'm very puzzled.

See the earlier response to the “Essential revisions” section.

Reviewer #2:[…] Unfortunately, their attempts at identifying the genetic regulators of the process were unsuccessful. The genetic screen using act8-mNeonGreen was inspired, even though it did not work. However, the remainder of the focused didn't seem as logical. While the choice of protein types (signaling, transcription, cell division and growth) made sense, the chosen genes seemed random. There were much more rational choices within these groups. DhkA is a developmentally regulated signaling protein that is known to be involved in dedifferentiation. ChdA, ChdB and ChdC are supposed chromatin remodeling proteins that are master regulators for developmental gene expression. There are multiple G α proteins that are known to be involved in food sensing during growth. Almost all of the mutants of all of these gene are able to reach the mound stage of development, so can be examined in this dedifferentiation system. While I am not suggesting that the authors perform these experiments in order to have this manuscript published, I would like a clearer rationale for the mutants that they did choose and why they are better choices than others.

These are interesting suggestions of gene classes. We checked the expression of the *chdA, chdB* and *chdC* genes in our RNAseq data and found they are strongly induced during development and rapidly repressed during dedifferentiation. These would not be candidates using our criteria. Of the G-α proteins, most showed no induction during dedifferentiation, although two of the genes did. However, the induction showed by these genes was not sufficient to satisfy our induction criteria, and the genes did not pass cutoff. We have expanded the rationale in the text for the selection of the genes we decided to test functionally.

I would also be wary of claiming that the dedifferentiation programme is genetically robust for the same reasons. Even though the multitude of mutants they tested didn't show large effects on dedifferentiation, they left out some fairly obvious ones. It would be safer to say that dedifferentiation is more genetically robust that differentiation, since there are a lot more genes that disrupt differentiation. However, this probably has more to do with the fact that development and differentiation requires a higher degree of temporal coordination than dedifferentiation. For dedifferentiation, it makes sense that it doesn't matter whether regulated cAR1 expression occurs before or after adenylyl cyclase expression or G α 2 expression. However, during development, the relative timing of expression is crucial.

Exactly our view: we have described this in the Discussion.

The cryS experiments were critical in demonstrating that cell fate was not responsible for the two different populations of dedifferentiated cells identified based on motility and division speed. I would ask that the control be mentioned showing proper CrS-mNeonGreen expression in the mounds before dissociation otherwise we don't know whether the system worked well-enough to identify actual prestalk cells. I'd also request a reanalysis of the data to see whether the high/low CryS-mNeonGreen expressing cells had the same ratios of high speed/low speed cells as the entire population. It may be that there is an enrichment of one type of cell based on CryS-mNeonGreeen expression. The data should already exist. It would just need to be analyzed differently.

Good questions. CryS is a robust prestalk marker in mounds and slugs- we have shown this using multiple approaches (transcriptomics and live imaging) in an earlier study (Antolovic et al., 2019). We carried out the reanalysis requested by the reviewer and found no difference in the motility distributions of the high and low CryS cells. The KS test p-values of this analysis have been incorporated into the text associated with Figure 6.